# Benefits of Silvopastoral Systems for Keeping Beef Cattle

**DOI:** 10.3390/ani11040992

**Published:** 2021-04-01

**Authors:** Stella Maris Huertas, Pablo Ernesto Bobadilla, Ignacio Alcántara, Emilie Akkermans, Frank J. C. M. van Eerdenburg

**Affiliations:** 1Departamento de Salud Publica Veterinaria, Facultad de Veterinaria, Universidad de la República Uruguay, Montevideo 11600, Uruguay; stellamaris32@gmail.com (S.M.H.); pabloe.bobadilla@gmail.com (P.E.B.); nalcann@gmail.com (I.A.); 2OIE Collaborating Centre for Animal Welfare and Livestock Production Systems for Uruguay—A multi-national OIE Centre, Universidad de la República Uruguay, Montevideo 11600, Uruguay; emilieakkermans@gmail.com; 3Department of Population Health Sciences-Farm Animal Health, Faculty of Veterinary Medicine, Utrecht University, Yalelaan 7, 3584 CL Utrecht, The Netherlands

**Keywords:** animal welfare, silvopastoral systems, open pasture, body weight, beef cattle

## Abstract

**Simple Summary:**

Silvopastoral Systems (SPS) are agroforestry arrangements that combine forage grasses with shrubs and trees for animal nutrition and complementary uses. Their contribution to sustainable livestock production and animal welfare is well recognized, mainly with Zebu cattle in tropical regions. In order to obtain information in temperate climate zones, 130 beef cattle of European breeds were selected from four commercial herds and randomly allocated to two contiguous plots: SPS and Open Pastures Systems (OPS). The trees in the SPS were *Eucalyptus globulus globulus* for paper pulp production. They were planted with two meters between each tree over native, diverse grasses. Over the course of one year, individual body weights and animal welfare indicators were collected every 45 days. There were no differences observed in weight gain between the two systems. No sign of impaired welfare, such as lameness, integument alterations, coughing, nasal/ocular discharge, or hampered respiration, was observed in either system. Silvopastoral systems offer animals a sustainable and richer environment, which will improve their welfare. The additional income provided by the wood production allows the farmers to maintain their traditional cattle farming lifestyle.

**Abstract:**

The potential benefits of keeping Zebu cattle in silvopastoral systems are well described in tropical regions. In order to obtain information on European breeds of beef cattle (*Bos taurus taurus*) in temperate climate zones, individual records of body weight and welfare indicators were obtained from 130 beef cattle. These belonged to four herds and were randomly allocated to two contiguous plots: Silvopastoral Systems (SPS) and Open Pastures Systems (OPS). The SPS in this study were areas with exotic trees of *Eucalyptus globulus globulus* for paper pulp production planted in a 2 × 2 design (two meters between each tree) over diverse, native grasses. The OPS were large open areas with a great diversity of native grasses, herbs, and small plots of trees where the animals could rest and shelter from extreme weather conditions. Over the course of one year, individual body weights and a number of specific animal welfare indicators were measured every 45 days. After a descriptive analysis, a generalized linear mixed model (GLMM) with a Gaussian distribution, with time and system (OPS or SPS) fitted as fixed effects and individuals nested by herd as random intercepts, was used. The results showed that weight gain did not differ between the two systems. None of the animals showed any sign of impaired welfare in either system over the study period. Silvopastoral systems offer animals a sustainable and richer environment that will improves their welfare. The additional income provided by the wood production allows the farmers to maintain their traditional cattle farming lifestyle.

## 1. Introduction

Silvopastoral Systems (SPS) are agroforestry arrangements that deliberately combine forage plants, such as grasses and legumes, with shrubs and trees for animal nutrition and complementary uses [1,2,3]. They contribute to sustainable livestock production, because they reduce the impact on natural resources and increase the efficiency and profitability of an area of land. Furthermore, food security and animal welfare are improved in SPS [4,5]. The integration of forestry and livestock production appears to be more productive, profitable, and sustainable than each separately [3,6]. Animal welfare is improved, because SPS provide shade and shelter for livestock during hot and sunny days [5] and allow animals to dedicate a greater percentage of time to grazing and rumination activities [3,7]. In studies in tropical regions with Zebu breeds, the cattle in SPS showed higher body condition than did those in pastures without trees. This measure is seen as a strong indicator of animal welfare [8]. Moreover, it is reported that animals in SPS have less anxiety and fear, associated with a greater possibility of partial or complete concealment [9]. Traditionally, in Uruguay, a country with a subtropical/temperate climate, European breeds of beef cattle (*Bos taurus taurus*) are kept on native Open Pastures Systems (OPS) for meat production [10]. Cattle nutrition in OPS in Uruguay is based on a great diversity of native grasses and herbs (more than 400 species), mainly composed of *Axonupus* spp., *Paspalum notatum* and *P. dilatatum*, *Piptochaetum stipoides*, *Cynodon dactylon*, and dwarf weeds [11], which differs from systems without trees and monocultures of grass species as described in Mancera’s studies with Zebu cattle in the tropics [8]. In recent decades, SPS have been introduced in Uruguay using an exotic tree species, *Eucalyptus* spp., as the main forestry component. This exotic tree genus is used for paper production and is well adapted to the country’s conditions. The planting design allows adequate passage of light through the “open” structure of the branches and leaves of *Eucalyptus* spp. [12], permitting sufficient growth of grasses to sustain livestock farming. Although Uruguay is located in the temperate climate zone of the planet, during the summer, temperatures often surpass 30 °C for several consecutive days [13]. When the environmental temperature and relative humidity rise, cattle need extra energy for cooling and eat less [6,14,15]. Shade is then essential. The temperature humidity index (THI), created by Thorn in 1959 [16], combines ambient temperature and relative humidity and is considered as “normal” below 74. Levels between 75 and 78 are treated as “mild”, those between 79 and 83 as “dangerous”, and measures over 84 as “emergency”. Trees are a good way to provide shade and thus diminish the risk of heat stress.

There are different ways of integrating forestry and livestock production in SPS worldwide, either in arrangements made by farmers or as a result of the adaptation and management of natural ecosystems [17,18]. The main forms of SPS are (i) scattered trees in pasturelands, (ii) timber plantations with livestock grazing areas, (iii) pastures between tree alleys, windbreaks, live fences, and fodder banks with shrubs, and (iv) intensive silvopastoral systems [6,10,19]. In Uruguay, cattle ranching is traditionally extensive in OPS, with a group of trees planted in the center of the open field to provide shade for the cows [10]. The main differences between temperate and tropical climates are in temperature and rainfall, which determine the growth capacity of different plant species [20].

Because SPS improve the microclimate for the animals during extreme weather, and because the wood produced by the trees will add to the income of the farmer, we designed this study to compare the welfare and productive performance of cattle from European breeds in SPS and OPS in Uruguay, a country with a temperate climate.

## 2. Materials and Methods

### 2.1. Study Site

The study was carried out in four locations of southeast Uruguay in South America, from February 2016 to February 2017.

### 2.2. Herds and Treatments

A total of 130 beef cattle of European breeds (*Bos taurus taurus*), from four commercial herds, were randomly assigned to two equal and contiguous plots: SPS and OPS, as summarized in Figure 1.

#### 2.2.1. Silvopastoral Systems (SPS)

The SPS studied in this research were areas with native grasses and *Eucalyptus globulus globulus* trees for paper pulp production. The trees were planted with a space of two meters between each tree (see Figure 2), besides the open spaces constituted by areas that could not be planted and fire-break roads [21].

The characteristics of the SPS under study are summarized in Table 1.

#### 2.2.2. Open Pastures Systems (OPS)

In our study, the OPS were large open areas with small plots of trees, generally older than those in the SPS and mainly there as places for animals to rest and shelter, as we can see in Figure 3.

#### 2.2.3. Additional Information

The rations given to the animals consisted entirely of native grass; no nutritional supplements were used. Dry matter (DM) analyses of the native grass were carried out by cutting samples of 200 g of grass at a height of around 5 cm in winter and 7 cm in summer, then drying them in an oven at 60–65 °C. The results are expressed as kilograms per hectare. The results showed averages of 280 kg/ha in winter and 680 kg/ha in summer for the SPS, and averages of 230 kg/ha in winter and 1255 kg/ha in summer for the OPS. These are low quantities when compared to those for other countries, but it should be noted that Uruguayan beef cattle farmers do not use fertilizers to stimulate grass growth. Furthermore, the number of animals per hectare is very low (0.8 animals/ha), which makes it similar to the natural habitat. Both the SPS and OPS had a natural water supply throughout the year.

### 2.3. Data Collection

Animals of all herds and systems were observed and assessed every 45 days for approximately one year, with the exception of Herd D (cows), where the visits were more spaced out as requested by the farmer in order to avoid additional handling stress closer to the end of pregnancy.

#### 2.3.1. Welfare Indicator Assessment

On visiting days, the animals were herded from the plots and left in separate pens for a few hours, without the presence of humans or movements that may cause stress.

After that time period, trained observers performed assessments of selected animal welfare indicators according to the Welfare Quality^®^ (WQ) protocol [22], adapted to local farming conditions [23]. Although, so far, no measure has been developed to evaluate the thermal comfort criterion in the WQ protocol, we consider it an important parameter for the welfare of the animals [14,24,25,26]. Non-hampered respiration is considered a measure of the absence of disease, as shown in Table 2. Therefore, the 130 animals, selected as described in Section 2.2 and Figure 1, were observed and considered positive if at least one animal breathed rapidly through the mouth (adapted from the panting scores proposed by Davis and Mader, 2003 [27]).

#### 2.3.2. Body Weight

All animals were periodically weighed, individually, on a scale (*Tru-Test*^®^ Model Eziweigh5/MP600; Almacen Rural, Montevideo, Uruguay) placed at the exit of the chute.

### 2.4. Records of Environmental Temperature and Humidity

Local weather information was obtained from the Uruguayan Agroclimatic data bank [13]. The variables collected were the daily maximum, minimum, and average values of the ambient temperature (°C) and relative humidity (%) for the study period. The temperature humidity index (THI) was calculated using the formula proposed by Thorn (1994) [16].

### 2.5. Statistical Analyses

In order to analyze the relationships between the response variable (weight, a numerical continuous one) and the explanatory variables (herd and system), an exploratory data analysis was performed using box-plot diagrams to visualize the weights by herd and system. The temporal dimension was also incorporated into the graphs in order to determine if there was an effect of individual evolution (at the animal and herd level) on weight gain. The information obtained from the descriptive graphs was used to select the best family of models and the most relevant variables in the context of the hypotheses set up [28].

Individual weight, as the response variable, was subjected to a generalized linear mixed model (GLMM) with a Gaussian distribution and identity as the link function, with time and system (OPS or SPS) fitted as fixed effects and individuals nested by herd as random intercepts. Additionally, the residual structure was adjusted to include the autocorrelated structure of the longitudinal design [29,30].

The model’s goodness of fit was assessed by the calculation of marginal (R^2^m) and conditional (R^2^c) coefficients of determination for generalized mixed-effect models [31]. For the data analysis, the statistical software R [32] was used. Graphics were constructed using the “*ggplot2*” package [33], models were fitted with the “*nlme*” package [34], and the R^2^ goodness of fit was estimated with the (r.squared GLMM) function implemented in the “MuMin” package [35].

## 3. Results

A total of 787 individual records of body weight and welfare indicators were obtained from 130 beef cattle belonging to Herds A, B, C, and D.

### 3.1. Welfare Indicator Assessment

All the animals observed scored a value of 0 for lameness and integument alterations, such as hairless patches and lesions/swellings. Furthermore, none of the animals had any sign of coughing, nasal discharge, ocular discharge, hampered respiration, diarrhea, or bloated rumen. Aggressive or abnormal behaviors were also not observed.

### 3.2. Live Body Weight

In Figure 4, the animal weight data in the four herds are summarized. Overall, there were no significant differences in weights between SPS and OPS within the same herds.

#### Generalized Mixed-Effect Model (GLMM)

The estimated marginal mean and 95% confidence intervals of weight by system and visit for the GLMM used are presented in Table 3. The results of the GLMM showed no differences between systems when the remaining explanatory variables of the experimental design were controlled. The analysis of the residuals showed that the chosen model was adequate. Figure 5 shows the mean weight values and their 95% confidence intervals. These were estimated by a GLMM for each of the eight visits. There were no differences between groups, and the magnitudes of the increase are shown in Table 3.

### 3.3. Records of Environmental Temperature and Humidity

The daily minimum, maximum, and average records of ambient temperature in degrees Celsius (AT), relative humidity as a percentage (RH), and temperature humidity index (THI) during the study period can be seen in Table 4. Note that in summer, fall, and spring, the maximum daily THI reached the emergency zone, while the daily average remained in the normal zone in all of the seasons.

## 4. Discussion

In the present study, we assessed several animal-based welfare indicators, and we did not find any animals that showed signs of poor welfare, in either the OPS or SPS. Apparently, an appropriate physical environment minimizes the risk of injury and diseases to animals, as suggested by Fraser et al. 2013 [36]. It is also accepted that environments that supply the needs of the animals result in good welfare [37]. According to Bouissou et al. [38], livestock adapt differently to different environments, and in crowded conditions, an animal cannot maintain sufficient individual distance and is forced to fight higher-ranking individuals. These agonistic interactions may result in injuries, which were not observed in the present study.

Furthermore, in extensive systems, animals can easily move away to avoid confrontation. There is a general opinion that extensive systems are always beneficial for animal welfare as they mimic the cattle’s natural habitat. The cattle can perform positive behaviors such as grazing and rumination [39,40]. This is probably the reason why we did not observe injured animals or agonistic behaviors. Neither did we observe signs of heat stress, which were reported by other authors in investigations that were carried out in tropical regions and with Zebu cattle [9,41]. However, some authors stated that animals in extensive systems do not always have acceptable welfare, because they may be more exposed to predators, extreme weather conditions, lack of quality food, and fewer staff to take care of them [37,42]. None of these factors were present in our study. In Uruguay, there are no predators for cattle, and the cattle can seek shelter under the trees during extreme weather. This is also true for OPS, as there are small plots of trees (Figure 3). Moreover, due to the low density of animals (0.8 animals per hectare), there was enough food available, as demonstrated in the growth rate.

It has been reported before that the inclusion of trees in SPS is considered to improve animal welfare, and it has been demonstrated to be better in various ways than pasture-only systems for animals [43]. The increase in biodiversity and benefits for farmers make SPS sustainable, where many other beef production systems are not [42]. Brazeiro et al. [44] found that the overall species richness in the landscape did not decrease with increasing forestation. The tree species *Eucalyptus globulus globulus* does not have a very dense leaf coverage, allowing sufficient light to pass through to the ground. Studies performed in Australia showed that pasture productivity (quality and quantity) was significantly higher under reasonably dense, mature, native eucalyptus cover. This might be due to an improved microclimate (higher winter temperatures and lower evaporation) and the higher soil organic matter content under the trees [45].

Regarding productive performance in our study, animals in the OPS and SPS had similar weights and weight gains, as well as similar body conditions, in contrast to findings by other authors that cattle in SPS in tropical regions had higher body condition than did cattle in grasslands without trees [8]. It should be noted that in the OPS in our study, the nutrition was based on a great diversity of native grasses and herbs [11], and the weather conditions were friendlier to animals because they could seek shelter under the trees, in contrast to the treeless monoculture systems of the previous findings [8].

With respect to body weight, the result of the present study differs from previous findings, where the annual weight gain per animal for dairy heifers was greater and body condition was higher in SPS [8,46]. In the present study, the body condition of the animals did not show an increase, but remained between 3 and 4 [47]. It should be noted that these studies were carried out in a tropical region and with SPS established over a monoculture of Brachiaria decumbens. Studies under similar conditions are scarce; nevertheless, in Uruguay, some authors working with European cattle breeds and *Eucalyptus* spp. found greater weight gains when the animals had access to shade. They found an improvement in grazing activity and daily weight gain in Hereford steers, without affecting feed intake [48,49]. However, none of these studies were conducted under conditions similar to those in the present study, so comparisons are difficult.

In tropical regions, the presence of forage trees in pastures enhances the yield and nutritional quality of food available for animal feeding. A combination of grazing species and trees was found to produce more meat per unit area per year [50]. Probably, the higher frequency of grass species with high nutritional value, even with low dry matter values, in both OPS and SPS in Uruguay can partly explain our results of good performance in terms of animal weight gain [51,52].

Although Uruguay’s livestock production has been undertaken in extensive systems on natural and diverse grasslands for hundreds of years, the growing demand for pulpwood has promoted an increase in SPS in areas less suitable for agriculture, as especially defined in the National forest law N^o^ 15,939 (1987). When compared with longer-rotation timber crops, pulpwood plantations have relatively lower input costs and require fewer management interventions, resulting in good financial returns in a short time. The valorization of intangible environmental benefits from agroforestry, such as efforts to mitigate climate change, water and biodiversity protection, erosion control, and shelter, may result in the emergence of markets for carbon credits and bioenergy [45]. This could result in extra income on top of the sale of wood.

Despite the fact that the daily THI maximum reached the emergency zone in summer, autumn, and spring, no animals with signs of heat stress, such as panting or shortness of breath, were observed. This could be partially explained by the cattle also having access to shade (small plots of trees) in the OPS under study. The SPS, however, also provided shade for cattle in the area where they grazed, so they could dedicate a greater percentage of time to grazing. Therefore, the THI thresholds, which are widely used for dairy cattle, may not be useful for beef cattle in OPS and SPS [53]. In addition, the presence of a permanent natural water supply in all paddocks will have contributed to mitigating potential heat stress [48]. Moreover, according to the NRC beef cattle feed intake guidelines, low-fiber diets appear to cause less heat stress than do high-fiber diets [39].

The welfare of humans and the welfare of animals are closely linked. Positive relations with animals are an important source of comfort, social contact, and cultural identification for many people. Furthermore, the One Welfare concept, to which we subscribe, recognizes the relationship between animal welfare, human welfare, and the environment [54].

## 5. Conclusions

The presence of *Eucalyptus* spp. planted on treeless grasslands in parts of Uruguay did not impair the performance or welfare of certain categories of European beef cattle breeds allocated to commercial plantations. Under local production conditions, welfare and productive performance (measured as weight gain) did not differ between the assessed systems. Silvopastoral systems offer animals a sustainable and richer environment, which will improve their welfare. Further, the additional income provided by the wood production allows the farmers to maintain their traditional cattle farming lifestyle.

## Figures and Tables

**Figure 1 animals-11-00992-f001:**
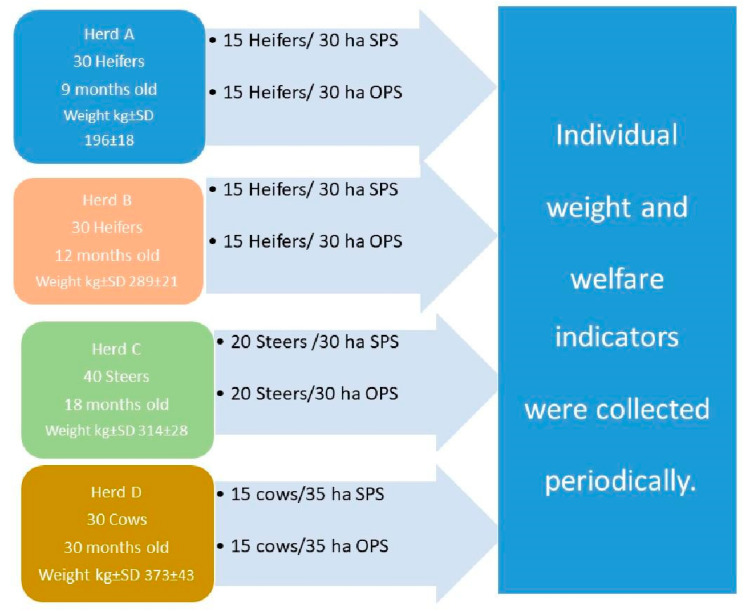
Main characteristics of the research design. Mean weight ± SD (SD: standard deviation) are given in kilograms. SPS, silvopastoral systems, OPS, open pasture systems. Note: Once assigned to each plot, animals were maintained there for the entire study period, and all animals were individually identified. A total of 130 animals were studied.

**Figure 2 animals-11-00992-f002:**
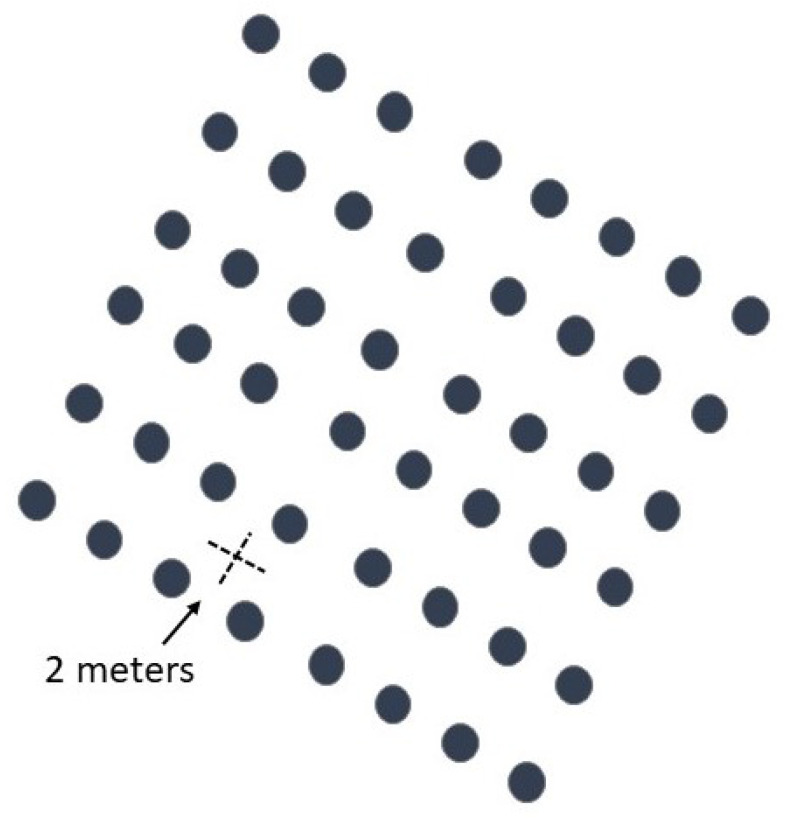
Scheme with a 2 × 2 tree plantation design.

**Figure 3 animals-11-00992-f003:**
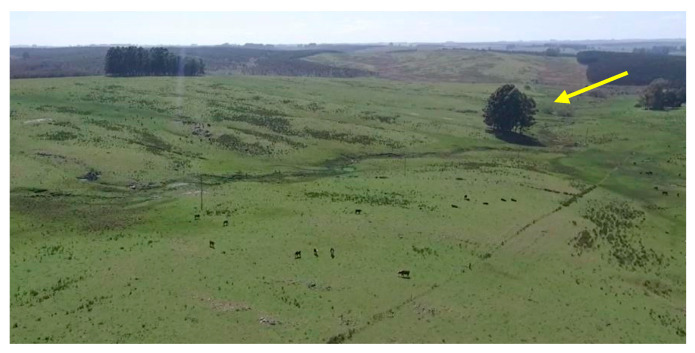
A picture of one of the open pasture systems (OPS) under study, showing a small plot of trees (see yellow arrow).

**Figure 4 animals-11-00992-f004:**
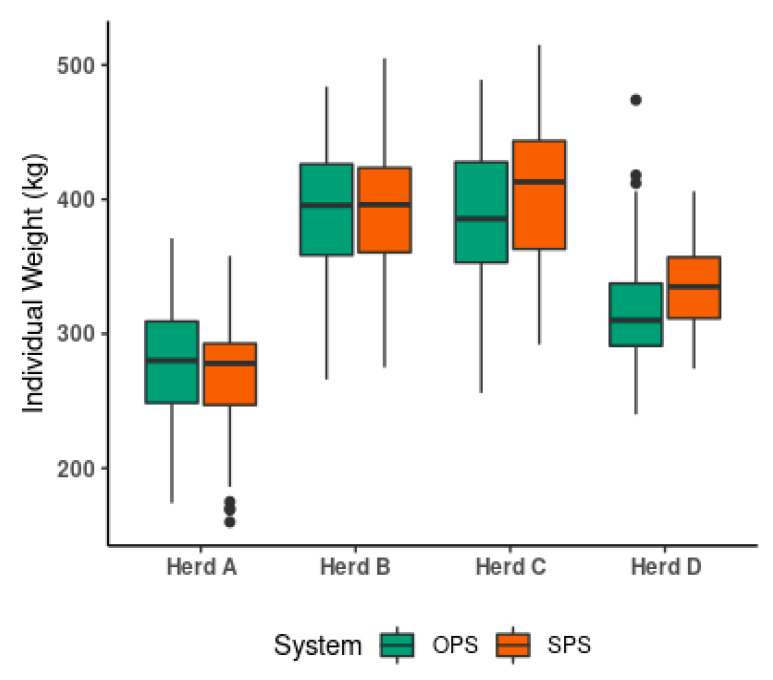
Distribution of individual weights (kg) expressed as percentile ranks for herds and systems. OPS, Open pasture systems; SPS, Silvopastoral systems. Note: The horizontal line in each box is the 50th percentile of the population (the median). Boxplots: boxes with interquartile range and median, whiskers with minimum and maximum dots with extreme values.

**Figure 5 animals-11-00992-f005:**
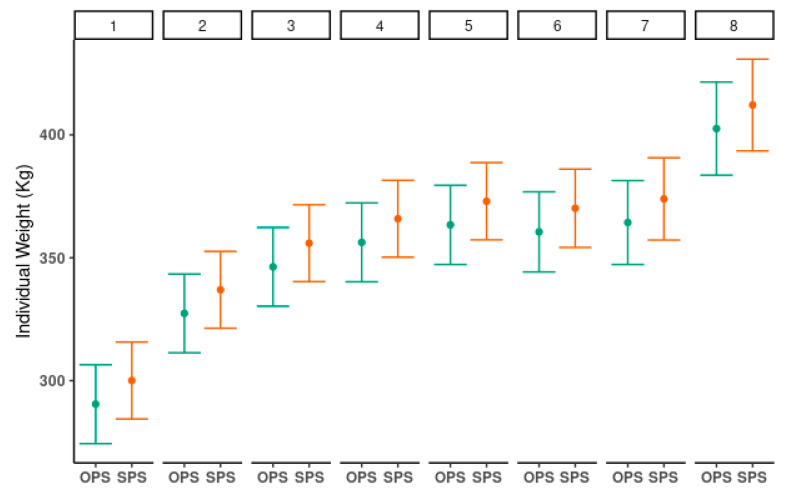
Mean weights and 95% confidence intervals of individual weights (kg) by herd and system across the eight visits. OPS: Open pastures systems; SPS: Silvopastoral systems.

**Table 1 animals-11-00992-t001:** Main characteristics of the Silvopastoral Systems (SPS) under study.

Characteristics of SPS	Type
*Eucalyptus* spp.	*globulus*
Average age of the trees (years old)	6
Design of plantation	2 × 2
Canopy level	Several layers, new shoots and harvest remains
Average Density (tree/ha)	830
Average DBH * (cm)	11.4
Average height (m)	11.7
Total volume (m^3^/ha)	60.5
Overall status	Acceptable health status

* DBH, diameter at breast height of the trees.

**Table 2 animals-11-00992-t002:** Some measures based on the Welfare Quality^®^ assessment protocol. Adapted from ref. [23].

Animal Welfare Indicator	Welfare Criterion	Measures
Good Feeding	Absence of prolonged hunger	Body condition score
Good health	Absence of injuries	Lameness, integument alterations
	Absence of disease	Coughing, nasal discharge, ocular discharge, hampered respiration, diarrhoea, bloated rumen, mortality
Appropriate behavior	Expression of social behaviors	Agonistic behaviors, cohesive behaviors

**Table 3 animals-11-00992-t003:** Generalized linear mixed model (GLMM) results of weight means by system across eight visits. The values are presented in absolute terms for Visit 1, and from Visit 2 onwards, only the increases with respect to the initial one are shown (which were the same for each group).

**Fixed Effects**		
Variable	Estimated weight ^a^	CI ^b^
systemOP: visit_1	290.5	(274.4; 306.5)
systemSPS: visit_1	300.1	(284.4; 315.7)
visit_2	36.9	(31.2; 42.6)
visit_3	55.9	(48.8; 62.9)
visit_4	65.8	(58.1; 73.5)
visit_5	72.9	(64.8; 81.1)
visit_6	70.1	(61.4; 78.7)
visit_7	73.9	(63.8; 84.0)
visit_8	112.1	(99.0; 125.1)
**Random Effects**		
Variable	Estimated SD	
herd/id ^c^	41.1 (Intercept); 33.2 (Residual)	
**Goodness of fit**		
R^2^m ^d^	15%	
R^2^c ^e^	79%	

^a^ Estimated weight: Mean individual weight (kg) estimated by the model; ^b^ CI: Confidence intervals (95%) for the estimated mean; ^c^ herd/id: Standard deviation of the nested mixed effects for the unique individuals by herd; ^d^ R^2^m: Marginal R^2^, represents the variance explained by the fixed effects; ^e^ R^2^c: Conditional R^2^, represents the variance explained by the entire model, including both fixed and random effects.

**Table 4 animals-11-00992-t004:** Average daily minimum and maximum ambient temperature in degrees Celsius (AT), relative humidity as a percentage (RH), and records of the temperature humidity index (THI) during the study period.

Season	Daily Minimum			Daily Maximum			Daily Average		
	AT	RH	THI	AT	RH	THI	AT	RH	THI
summer	10	65	51.6	35	98	94.9	23.3	81	72.5
fall	−1	74	34.2	32	98	89.6	17.3	88.6	63
winter	0	71	36.2	20	98	68.1	10.7	87.6	51.9
spring	6	57	46.5	34	96	92.8	18.4	76.5	64.4

AT: Ambient temperature in °C; RH: Relative humidity in %; THI: Temperature humidity index.

## Data Availability

No new data were created or analyzed in this study. Data sharing is not applicable to this article.

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
