# Peer review of "Benefits of Silvopastoral Systems for Keeping Beef Cattle"

_animals, 2021, doi:10.3390/ani11040992_

Round 1

Reviewer 1 Report

Comments to authors

Overall: The authors need to improve the English grammar, word choice, and word order in this paper.

L60     What are some of the major native grass, legumes and herbs present in these pastures?

L 83     Word order, photosynthesis before growth. This can be argued but I think in context to the paragraph reversing the order would be better.

Figure 1     Round the numbers presented to nearest unit. The decimal portion is not needed and is not relevant. Is the number to the righthand side of the “±” sign a standard deviation or a standard error term? Tell the reader which it is. Is the number a body weight in kg? Give the units.

Table 2     Use either a “,” or “.” to represent a decimal but do not intermix the use of these two symbols. Are the decimal values relevant and needed? I suggest rounding to nearest unit. What does “Dirty forest” mean?

L119-122     Forage dry matter available at 230 to 680 kg/ha is extremely low, if measured at ground level. Is this forage available above a defined grazing height? This part of the methods needs to be described in detail.

L183-187     This discussion of body weight reflects animal assignment to treatments not treatment effects, as alluded to in the statement. Remove “ In spite of the weight distribution overlap observed for both systems within herds, those C and D showed heavier animals in SPS than those of OPS, additionally we can observe that weights in SPS for herds A and D showed less dispersion (more compact boxes) when compared with their counterparts in OPS.” lines from the text.

L194-195     Likewise remove L194-195 “However, the SPS always showed higher 194 weight values than OPS.”  which makes the same mistake as in L183-187. There is a difference between weight and weight gain.

Figure 4.     Next to the “System” marker the identifier should be “OPS” and both OPS and SPS should be defined in the text describing the figure.

L220     Just a note: low fiber diets appear to cause less heat stress than do high fiber diets according to the NRC beef feed intake guidelines.

Table 3.     This is a confusing, hard to interpret table. What does it mean to those who are not users of the R statistical packages? I suggest modifying the table. In the “herd/id” is the “Residual” a standard deviation?

Figure 5.     This is an excellent presentation of the animal weight data. (Possibly same as in Table 3?). Do use “OPS” on the X-axis as defined in the label text describing the figure.

L243     The authors state there was ample food. Define how available forage mass was measured. The numbers, as presented, imply extremely low available forage mass to the point where animals are starving. If this is forage mass above a 10 cm grazing height it is adequate for these classes of cattle. Define the methods used to measure available forage to justify the statement of “ample food”.

L257-258     Drop the sentence “Despite the lack of statistical significance, it should be noted that for every visit, animals in SPS had higher weights.”  This statement it is not meaningful and gives an incorrect interpretation of the data.

L267     Report what the body condition score of the cattle was over the course of the study.

L280     State “animal weight gain” not “animal weight”.

L302     Is this a typographical mistake? Should “wher the grazed” be “where they grazed”?

Author Response

Dear reviewer 1,

We appreciate the time you have dedicated to our manuscript "animals-1101382". We have improved English grammar, word selection and order. We gladly accepted your comments and suggestions and we have worked trying to improve concepts.

The details are as follows:

Your comment was: L60 “ What are some of the major native grass, legumes and herbs present in these pastures?”

  • We add examples: “mainly composed of Axonupus spp., Paspalum notatum and P. dilatatum, Piptochaetum stipoides, Cynodon dactylon and dwarf weeds.”(see L64-65)

Your comment was: L83.  “Word order, photosynthesis before growth. This can be argued but I think in context to the paragraph reversing the order would be better”.

  • We have arranged the following paragraph. “In Uruguay, cattle ranching is traditionally extensive in OPS, with the addition of a group of trees, planted in the center of the field, for the provision of shade for the cows and production of timber, which is also economically important for the farmers [10]. It is known that the main difference between temperate and tropical climate is the variation in temperatures and rainfall, which determines photosynthesis and thus the growth capacity of different plant species [21]”(see l84-88)

Your comment was: Figure 1 “Round the numbers presented to nearest unit. The decimal portion is not needed and is not relevant. Is the number to the right and side of the “±” sign a standard deviation or a standard error term? Tell the reader which it is. Is the number a body weight in kg? Give the units”.

  • We have rounded the numbers and corrected the units.

Your comment was: Table 2 “Use either a “,” or “.” to represent a decimal but do not intermix the use of these two symbols. Are the decimal values relevant and needed? I suggest rounding to nearest unit. What does “Dirty forest” mean?”

  • We have changed “comma” for “dot” and we deleted “Dirty forest” because is what remains after harvest, so it was the same.

Your comment was: L119-122. “Forage dry matter available at 230 to 680 kg/ha is extremely low, if measured at ground level. Is this forage available above a defined grazing height? This part of the methods needs to be described in detail”.

  • We added more information related to this topic: “Animals were fed entirely on native grass; no nutritional supplements were used. Dry matter (DM) analyses of native grass were carried out by cutting samples of 200 grams of grass that height in winter around 5 cm and in autumn 7 cm, drying them in an oven at 60-65oC and expressed as kg/ha. The results showed an average of 280kg/ha in winter and 680 kg/ha in summer for SPS; and an average of 230kg/ha in winter and 1255 kg/ha in summer for OPS. These are low quantities when compared to other countries, but it should be noted that Uruguayan beef cattle farmers do not use fertilizers to stimulate grass growth. Besides, the number of animals per hectare is very low (0.8 animals/ha). It is thus the natural situation. Both SPS and OPS had a natural water supply throughout the year” (see L124-133).

Your comment was: L183-187. “This discussion of body weight reflects animal assignment to treatments not treatment effects, as alluded to in the statement. Remove “In spite of the weight distribution overlap observed for both systems within herds, those C and D showed heavier animals in SPS than those of OPS, additionally we can observe that weights in SPS for herds A and D showed less dispersion (more compact boxes) when compared with their counterparts in OPS.” lines from the text”

  • We have removed: “In spite of the weight distribution overlap observed for both systems within herds, those C and D showed heavier animals in SPS than those of OPS, additionally we can observe that weights in SPS for herds A and D showed less dispersion (more compact boxes) when compared with their counterparts in OPS.” (see 199-203)

Your comment was: L194-195 “Likewise remove L194-195 “However, the SPS always showed higher 194 weight values than OPS.”  which makes the same mistake as in L183-187. There is a difference between weight and weight gain”.

  • We have removed: “However, the SPS always showed higher 194 weight values than OPS.”(see L210-211)

Figure 4 ” Next to the “System” marker the identifier should be “OPS” and both OPS and SPS should be defined in the text describing the figure”.

  • We have changed into OPS

Your comment was: L220  “Just a note: low fiber diets appear to cause less heat stress than do high fiber diets according to the NRC beef feed intake guidelines”. 

  • We appreciate the comment and add: “In addition, according to the NRC beef feed intake guidelines, low fiber diets appear to cause less heat stress than do high fiber diets [40]” (see L234-235).

Table 3  “This is a confusing, hard to interpret table. What does it mean to those who are not users of the R statistical packages? I suggest modifying the table. In the “herd/id” is the “Residual” a standard deviation?”

  • We can understand that the table can be confused. We have tried to clarified the indicators.

In the case of the fixed-effects models (e.g., regression, ANOVA, general linear models), there is only one source of random variability. This source of variance is the random sample we take to measure our variables. In mixed effects models are different in that there is more than one source of random variability in the data. We account for these differences through the incorporation of random effects. In our case, random intercepts allow the outcome to be higher or lower for each animal in each heard (nested effects represented by herd/id). This allows us to make “broad level” inferences about the larger population of animals, which do not depend on a particular herd. In other words, we can now incorporate individual and herd variability in weight gain and improve our ability to describe how fixed effects relate to outcomes.  We can also talk directly about the variability of random effects, similar to how we talk about residual variance in linear models. Indeed, the variance in random factor tells you how much variability there is between individuals across all treatments.

Your comment was: Figure 5  “This is an excellent presentation of the animal weight data. (Possibly same as in Table 3?). Do use “OPS” on the X-axis as defined in the label text describing the figure.

  • We have changed into “OPS”.

Your comment was: L243 “The authors state there was ample food. Define how available forage mass was measured. The numbers, as presented, imply extremely low available forage mass to the point where animals are starving. If this is forage mass above a 10 cm grazing height it is adequate for these classes of cattle. Define the methods used to measure available forage to justify the statement of “ample food”.

  • We explained the Dry matter analysis in Mat & Mets. (see L129-136) and we have added: “Because of the low density of the animals, 0.8 animals per hectare, there was indeed enough to eat” (L264-265)

Your comment was: L257-258 “Drop the sentence “Despite the lack of statistical significance, it should be noted that for every visit, animals in SPS had higher weights.”  This statement it is not meaningful and gives an incorrect interpretation of the data”

  • We dropped the paragraph: see L279-281.

Your comment was: L267 “Report what the body condition score of the cattle was over the course of the study”.

  • We have added: ”In the present study, the body condition of the animals did not show an ostensible increase, remaining acceptable (between 3 and 4) [58]” (see L289-290)

Your comment was: L280  “State “animal weight gain” not “animal weight”. We have corrected to: “animal weight gain” (see L303)

Your comment was: L302 “Is this a typographical mistake? Should “wher the grazed” be “where they grazed”? we have corrected (see L 326)

Reviewer 2 Report

It is important to address the aspects of the paper also under temperate climate zones, so in this sense the paper is interesting. However, I miss a well-developed hypothesis. Why do you expect beneficial productive performance of the livestock, when trees are included to compete with the pasture???

Experimental design is fine.

The discussion can be shortened very much – concentrating on the interpretation of the experimental results only.  

Medio line 21 – 22. Statement should be omitted

Line 30. It is not appropriate to mention the number of individual animal records in the abstract

Line 35: I think there should not be a punctuation before ‘While’

Line 62 Uruguay !

Table 2: DHB and volume. Use dot for comma. What is meant by ‘dirty forest’; please explain.

Line 119. Animals were fed…

Section 2.2.3. How were available forage DM determined. Should probably be given by each farm  

Line 194. This statement seems not in accordance with the results in figure 4 – I know that there is referred to Figure 5, but it is clear from figure 5 that this difference were present from the very beginning of the experiment - anyway if not statistical significant I suggest that this should not be mentioned. I also suggest figure 5 to go out – does not really add to the paper.  

Table 3. What are the units? I suspect the slope values to reflect live weight gain.  

Lines 218-220: this sentence seems not relevant here – does not support the interpretation of results  

Line 251 -255. I think this information should be the basis for development of the hypothesis in the paper. As is the hypothesis is not convincing and should be improved

Line 257-258. See my previous comment to this statement

Line 194. This statement seems not in accordance with the results in figure 4 – I know that there is referred to Figure 5, but it is clear from figure 5 that this difference were present from the very beginning of the experiment - anyway if not statistical significant I suggest that this should not be mentioned. I also suggest figure 5 to go out – does not really add to the paper.  

Table 3. What are the units? I suspect the slope values to reflect live weight gain.  

Lines 218-220: this sentence seems not relevant here – does not support the interpretation of results  

Line 251 -255. I think this information should be the basis for development of the hypothesis in the paper. As is the hypothesis is not convincing and should be improved

Line 257-258. See my previous comment to this statement

Author Response

Dear reviewer 2,

We appreciate the time you have dedicated to our manuscript "animals-1101382" and we recognized very much that you consider interesting our work. We gladly accepted your comments and suggestions and we have worked trying to improve concepts.

The details are as follows:

Your comment was: L 21 – 22. “Statement should be omitted”.

  • We have deleted: “Results show that both systems do not differ statistically over the study time, however, animals at the SPS always had higher weight gains than at OPS.” And since it is one of the major findings of our study, we added: “There were no differences observed in weight gain in both systems.” (see L23)

Your comment was: L30. “It is not appropriate to mention the number of individual animal records in the abstract”

  • We have removed individual animal records of the abstract as suggested. We have removed: “a total of 787” (see L31)

Your comment was: Line 35: “I think there should not be a punctuation before ‘While’”.

  • We have deleted de “dot” and added a “comma” (see L35)

Your comment was: Line 62 Uruguay !

  • We have corrected the spelling of “Uruguay” (see L 64)

Your comment was: Table 2: “DHB and volume. Use dot for comma. What is meant by ‘dirty forest’; please explain”.

  • We have changed ”dot” for “comma” in values of DHB and volume; and we deleted “Dirty forest” because is what remains after harvest, so it was the same.

Your comment was: Line 119. “Animals were fed…”.

  • We have arranged to: “Animals were fed” (L 130)

Your comment was: Section 2.2.3. “How were available forage DM determined. Should probably be given by each farm”.

  • We added more information related to this topic. The results of DM were an average of each kind of system. “Animals were fed entirely on native grass, no nutritional supplements were used. Dry matter (DM) analyses of native grass were carried out by cutting samples of 200 grams of grass that height in winter around 5 cm and in autumn 7 cm, drying them in an oven at 60-65oC and expressed as kg/ha. The results showed an average of 280kg/ha in winter and 680 kg/ha in summer for SPS; and an average of 230kg/ha in winter and 1255 kg/ha in summer for OPS. These are low quantities when compared to other countries, but it should be noted that Uruguayan beef cattle farmers do not use fertilizers to stimulate grass growth. Besides, the number of animals per hectare is very low (0.8 animals/ha). It is thus the natural situation. Both SPS and OPS had a natural water supply throughout the year” (see L130-138).

Your comment was: Line 194. “This statement seems not in accordance with the results in figure 4 – I know that there is referred to Figure 5, but it is clear from figure 5 that this difference was present from the very beginning of the experiment - anyway if not statistical significant I suggest that this should not be mentioned. I also suggest figure 5 to go out – does not really add to the paper”. 

  • We agreed that it could be confusing, so we deleted the paragraph: “In spite of the weight distribution overlap observed for both systems within herds, those C and D showed heavier animals in SPS than those of OPS, additionally we can observe that weights in SPS for herds A and D showed less dispersion (more compact boxes) when compared with their counterparts in OPS” (see L 199-203), and we have located the Figure 4 immediately below.
  • We also moved forward the sentence: “Figure 5 shows the mean weight values and their 95% confidence intervals. These are the estimated by GLMM for each of the eight visits. There are no differences between groups and the magnitudes of the increase are shown in Tabe 3.” before Figure 5. We consider that this graphic can help to see the weight evolution, so we would prefer to keep it in the paper (see L 228-230). Furthermore, reviewer 1 liked this figure.
  • We then placed Table 3 with more explanations in order to facilitate the understanding of the results (see L219-226).

Your comment was: Table 3. “What are the units? I suspect the slope values to reflect live weight gain”. 

  • We have added some explanations to the Table 3 as suggested.

Your comment was: Lines 218-220: “this sentence seems not relevant here – does not support the interpretation of results”.

  • We accept that this sentence could be not relevant and we deleted: “Intensive systems can have poor floor conditions, leading to lameness and the animals may be fed with high starch and low fiber content diets which increases the risk of metabolic acidosis and heat stress in the summer season. We did not find any indications for these in the present study” (see L249-253)

And we added: “In addition, according to the NRC beef feed intake guidelines, low fiber diets appear to cause less heat stress than do high fiber diets [40]” (see L253-255).

Your comment was: Line 251 -255. “I think this information should be the basis for development of the hypothesis in the paper. As is the hypothesis is not convincing and should be improved”.

  • We thank you for this comment and we improved the hypothesis: “Because SPS provide a beneficial microclimate for the animals during extreme weather and the trees will add to the income of the farmer, we designed this study to compare the welfare and the productive performance of cattle from European breeds in SPS and OPS in Uruguay, a country with a temperate climate”. (see L93-97)

Your comment was: Line 257-258. “See my previous comment to this statement”.

  • We deleted: “Despite the lack of statistical significance, it should be noted that for every visit, animals in SPS had higher weights” (see 287-289)

Round 2

Reviewer 2 Report

I think the paper is now acceptable for publication 

Author Response

We thank the reviewer for his/her efforts